# Quantitative Analysis of Bone, Blood Vessels, and Metastases in Mice Tibiae Using Synchrotron Radiation Micro-Computed Tomography

**DOI:** 10.3390/cancers15235609

**Published:** 2023-11-28

**Authors:** Hao Xu, Max Langer

**Affiliations:** 1Institute of Innovation Science and Technology, Shenyang University, Dadong District, Wanghua South Street No. 21, Shenyang 110044, China; 2Université Grenoble Alpes, CNRS, UMR 5525, VetAgro Sup, Grenoble INP, TIMC, 38000 Grenoble, France; max.langer@univ-grenoble-alpes.fr

**Keywords:** synchrotron radiation microcomputed tomography, breast cancer bone metastases, anti-angiogenic drugs, metastases segmentation

## Abstract

**Simple Summary:**

Breast cancer bone metastasis is dangerous and common worldwide. Early diagnosis and treatment are important for increasing the survival rates of patients. The aim of this study was to quantify bone, blood vessels, and metastases in a murine model of breast cancer bone metastasis. We simultaneously segmented bone, blood vessels, and metastases in three-dimensional synchrotron radiation micro-computed tomography images. The following quantitative parameter measurements and statistical analysis demonstrated that a combined anti-angiogenic treatment significantly decreased the volume and thickness of blood vessels close to metastases. The preliminary results of this study can be further used to both gain a better understanding of interactions between blood vessels and metastases as well as to study the anti-angiogenic drug involvement in bone metastatic processes.

**Abstract:**

Bone metastases are one of the most dangerous consequences of breast cancer. Early diagnosis and treatment would slow down the development of the disease and increase the survival rates of patients. Bone micro-vasculature is believed to play a major role in the development of bone metastases. It could be used for both diagnosis and as a therapeutic target. Synchrotron radiation micro-computed tomography (SR-µCT) with a contrast agent of blood vessels has been used to analyze the bone vasculature both in healthy and in metastatic bone. However, few studies have investigated the local features of blood vessels around metastases so far. For this purpose, the metastases first need to be automatically segmented. This is a challenging task, however, since the metastases do not contribute a specific contrast to the three-dimensional (3D) SR-µCT images. Here, we propose a new method for the simultaneous segmentation of bone, blood vessels, and metastases from contrast enhanced 3D SR-µCT images based on the nnU-Net architecture. In this study, we showed that only minimal training data was required to achieve a high quality of segmentation. The proposed method allowed for the automatic segmentation of metastases and provided an improved segmentation of bone and blood vessels compared to previous methods while being much more efficient to apply once trained. Further, the automatic segmentation allowed for the measurement of vascular metastases interdistance and to restrict measurements to volumes of interest around the metastases. Finally, we quantitatively analyzed blood vessel parameters locally around metastases. This allowed for the demonstration that a combined anti-angiogenic treatment significantly decreased the volume and thickness of blood vessels close to metastases. The proposed method showed the capacity of the method to reveal new aspects of the blood vessel structure interaction with metastases. This could be further used to both define new targets for precocious detection of metastases as well as to study the kinetics of metastasis development in bone and the action of drugs on this process.

## 1. Introduction

Breast cancer is common in women worldwide, and bone metastasis is one of the most dangerous consequences of breast cancer. Breast cancer bone metastases can lead to poor prognosis and result in the death of patients [1,2,3]. Vessel development is believed to play a major role in the development of metastases [4,5]. Unfortunately, in the bone metastatic process, the mechanism of bone vasculature in response to tumor invasion is still poorly known at the three-dimensional (3D) organ level [6]. Thus, the quantitative analysis of bone, blood vessels, and metastases is required [7]. In particular, few studies have investigated the local features of blood vessels around metastases so far.

Imaging micro-vasculature in bone remains challenging due to the embedding of the vessels in the dense bone matrix. Histology with immunostaining and microscopy is a tool of choice to assess angiogenesis. However, only 2D planar information is provided for assessing blood vessels [8,9]. Laser confocal scanning microscopy (LCSM) has commonly been used to image bone vascularization with staining and provide 3D reconstructions. Yet, it is limited by the field of view and depth of penetration [10]. 3D X-ray micro-computed tomography (µCT) is a powerful tool to image vascularization within the whole bone with a contrast agent [11,12]. However, 3D X-ray µCT usually requires bone decalcification to visualize the vascular architecture. Until recently, research showed that it was possible to simultaneously visualize bone and blood vessels without decalcification using the polymer-based contrast agent µAngiofil and 3D X-ray μCT [13]. Synchrotron radiation is an emitted electromagnetic radiation in a wide spectral range from infrared to hard X-rays. A synchrotron radiation X-ray beam is characterized by the high flux and high degree of mono-chromaticity, compared to the standard X-ray [14,15,16]. Therefore, unlike conventional 3D X-ray µCT, synchrotron radiation µCT (SR-µCT) can distinguish the attenuation values of contrasted blood vessels and bone clearly. Thus, SR-µCT has allowed for the simultaneous visualization of 3D bone microstructures and vascular networks [17]. In addition, SR-µCT achieves higher spatial resolution and signal-to-noise ratio in the 3D images compared to standard µCT due to the high photon flux of the synchrotron source [18]. Electron storage rings are used to provide synchrotron radiation in dedicated facilities around world, such as ESRF (Europe), ALS (America), Spring-8 (Japan), SSRF (China). SR-µCT has been previously applied in rat models [17,19] and mice [7,20,21,22].

Previously, we developed a protocol to image and analyze bone and blood vessels in mouse bone using contrast agent perfusion, SR-µCT, and image segmentation based on region growing and a monogenic signal phase asymmetry controlled watershed [7,17,22]. However, some false positives remained for the blood vessels and the bone compartments. In addition, the computing cost was relatively high for 3D images with a large size. To analyze the relationship between metastases and vascularization, it is desirable to also automatically segment metastases. Segmenting metastases from these kinds of images is challenging, however, since the metastases are characterized only by missing bone structure in the µCT images; there is no contrast specifically from the metastases in these kinds of images. Designing a classical image analysis algorithm for this problem therefore seems very difficult, if not impossible.

We hypothesize that using machine learning would make it feasible to detect these missing structures. Therefore, we propose here to use a deep-learning based method to segment bone, blood vessels, and metastases simultaneously from 3D SR-µCT images with vessel contrast agent. A nnU-Net [23] is trained using 3D images where bone and vessels are segmented using our previous method [22] with manual corrections, and metastases are manually segmented. Generating training data for these kinds of 3D images with large size is costly and time consuming. We show that with our approach, only minimal training data is necessary to achieve good segmentation. The proposed method allowed for the automatic segmentation of metastases, and provided improved segmentation of bone and vessels compared to previous methods, while being much more efficient to apply once trained. Further, the automatic segmentation allowed for the measurement of vascular metastases interdistance and restriction to the volume of interest around the metastases. Finally, we quantitatively analyzed blood vessel parameters locally around metastases. This permitted the demonstration that a combined anti-angiogenic treatment significantly decreased the volume and thickness of blood vessels close to metastases.

## 2. Materials and Methods

### 2.1. Sample Preparation

Mice tibiae specimens used within this study were previously described in detail [7]. A sample set of 73 tibiae was collected from female mice which had been injected with luciferase-expressing human B02 breast cancer cells, placebo (P), and different anti-angiogenic drugs (Bevacizumab (B), Vatalanib (V) and a combination of both drugs (C)). In this study, two time points of 8 days (T1) and 22 days (T2) after tumor cell injection were set up (T1: 45 samples, T2: 28 samples). At T2, there were 7 samples per group for T2P, T2B, T2V, and T2C. The mice were sacrificed by anesthetic overdose and injected with a contrast agent (barium sulfate) for vascular imaging. Afterwards, tibiae were dissected, fixed in paraformaldehyde, embedded in methylmethacrylate, and kept for SR-µCT imaging.

Ethical approval for the protocol (DR2015-18) was provided by the ethics committee (CNREEA C2EA-55) andthe Minister of Higher Education, and Research and Innovation (Ministère français de l’Enseignement supérieur, de la Recherche et de l’Innovation, approval number: 2015121515281004).

Imaging experiments performed at the European Synchrotron Radiation Facility, Grenoble, France, on the ID19 beamline within this study have been previously described in detail [7,22]. In this study, we selected a VOI (volume of interest, 700×700×800) to segment bone, blood vessels, and metastases (voxel size: 3.5 µm).

### 2.2. Generation of the Training Data

For the training of the neural network, some data must be segmented into bone, blood vessels, metastases, and the background. In a previous study, we have segmented bone and blood vessels in 3D SR-µCT images using a monogenic signal phase-based watershed algorithm [22]. This yielded high quality segmentation, which was validated using synthetic and experimental data sets. There were still some false positives for blood vessel and bone compartments, however, due to partial volume effect and intensity variations in the SR-µCT images, as shown in Figure 1. Therefore, to improve the efficiency of the generation of the annotated training data, instead of manually segmenting the 3D images, an extremely time consuming task, a set of automatically segmented images was corrected for minor segmentation errors in blood vessels and bone by hand. The metastases, on the other hand, were segmented manually since no automatic algorithm is available for this task. Finally, we combined annotated bone, blood vessels, and metastases as the data set to train the nnU-Net network.

To quantify how much training data is necessary for yielding high quality segmentation, we generated 5, 10, and 15 VOIs (700×700×100) for training the nnU-Net network.

### 2.3. nn-Unet for Image Segmentation

The nnU-Net architecture [23] was used for the automatic segmentation of bone, blood vessels, and metastases in the SR-µCT images. Being wrapped around the standard U-Net architecture, the nnU-Net framework automatically designs and executes a network training pipeline to achieve competitive performance in segmentation tasks. The automatic configuration covers the entire segmentation pipeline, including preprocessing, network architecture, training, and post-processing for the arbitrary training database. As can be seen in Figure 2, three-dimensional SR-µCT images are input into the model and undergo a series of plain convolutions (Conv), instance normalization (IN), followed by a leaky rectified linear unit (lReLU) activation function in each computational block. Strided convolutions are used for downsampling, and upsampling is done with transposed convolutions. At each downsampling step the number of feature channels is increased, and conversely decreased at each upsampling step. Skip connections (concatenation) are used to add extra features from the encoder side of the network to the decoder side of the network. At the final layer, a 1×1×1 convolution (unpadded) is used to output segmentation map with the desired number of classes, as shown in Figure 2. As for data augmentation, rotations, scaling, Gaussian noise, Gaussian blur, brightness, contrast, simulation of low resolution, gamma correction, and mirroring were applied by the nnU-Net architecture. 

The networks were trained for 1000 epochs, in which one epoch is defined as 250 training iterations. All nnU-Net architectures were trained in a five-fold cross-validation [23]. An initial learning rate of 0.01 and a Nesterov momentum of 0.99 were used in the stochastic gradient descent optimization. The sum of the cross-entropy loss and the Dice loss were used as loss function to train the nnU-Net.

### 2.4. Evaluation Criteria

To evaluate segmentation quality, the Dice coefficient was measured to assess spatial overlap between the segmentation from the neural network and the manually prepared segmentation [24]. The Dice coefficient is given by:$$Dice=\frac{2\left|A\cap B\right|}{\left|A\right|+\left|B\right|}$$
where $\left|A\right|$ and $\left|B\right|$ represent the segmentation and ground truth, respectively. To calculate the Dice coefficient, we used the open source RStudio software 2022.12.0 [25] with programming language R [26].

### 2.5. Quantitative Parameter Extraction

The quantitative 3D parameters of the bone, blood vessels, and metastases were then calculated. The total volume (TV) of the sample was defined as the volume inside the outer contour of the bone. TV, bone volume (BV), vessel volume (VV), and metastases volume (Me.V) were measured by counting voxels in the corresponding compartments. The volume-based fractions, such as the normalized ratios BV/TV, VV/TV, and Me.V/TV were then calculated. In addition, the local vessel thickness was also measured and its average V.Th was reported. Finally, the vascular metastases interdistance (VMI), which was defined as the average of the local distance from each point in the blood vessel compartment to the metastasis compartment [17], was calculated. The parameters were calculated in a shell script with the Insight Toolkit (ITK), which is an open source library for image analysis [27,28].

### 2.6. Statistical Analysis

Hypothesis testing was performed on measured BV/TV, VV/TV, V.Th, and Me.V/TV between the T1 and T2 groups. It was also performed on the local VV/TV and V.Th around metastases (<700 µm) between the different drug treatments at T2. The Lilliefors test was used to test the normality of the data in each group. In the case of normal distribution and equal variance, the ANOVA F-test was used to test if there was a significant difference between the groups. We considered the difference statistically significant at the *p* < 0.05 level, *p* < 0.01 level, and *p* < 0.001 level. Statistical analysis was implemented using RStudio software.

## 3. Results

### 3.1. Qualitative Results of Segmentation

Example segmentation results are shown in Figure 3. (A) Three orthogonal slices of the original 3D volume. (B) The segmentation result of the corresponding slices. Qualitatively, the bone (white), vessels with contrast agent (red), and bone metastases (yellow) seem well segmented. 3D volume rendering of segmentation using the proposed method is shown as Figure 4, in which bone, blood vessels, and metastases are given in white, red and yellow, respectively. We used the open source image processing software Fiji/ImageJ 1.53c to perform the 3D volume rendering [29,30,31,32].

### 3.2. The Minimum Sample Size Required to Train a Model

To quantify how much training data is necessary for yielding high quality segmentation, we compared models trained by different sizes (5, 10, 15) of data sets. 12 representative original volumes were randomly selected as a test data set. Dice coefficients corresponding to vessel, bone, and metastases compartments were calculated separately as shown in Figure 5. With regard to the segmentation of bone and skeletal vasculature, the relatively small variances and high mean values on each group indicate that only 5 samples are required to achieve a high quality of prediction. However, as for metastases segmentation, there is a large variation in the group using 5 samples to train a model. It means that there are difficulties in predicting bone metastases correctly in some cases, possibly because the metastases were only characterized by missing bone structure in our images. In summary, to simultaneously segment bone, skeletal vasculature, and bone metastases with a relatively high segmentation quality, a minimal training data size of 10 is required.

### 3.3. Quantitative Analysis

When analyzing the bone, we found a significant increase in the BV/TV (*p* < 0.001) at T2 (Figure 6A). With regard to the blood vessels, at T2, VV/TV was significantly lower (*p* < 0.01) and V.Th was significantly higher (*p* < 0.01) than at T1 (Figure 6B,C). This is possibly due to the fact that the mice were of a young age (T1) and growing fast (T2). Specifically, the lower VV/TV at T2 might be mainly due to the large increase of TV. In addition, the Me.V/TV was significantly higher (*p* < 0.001) at T2 (Figure 6D). This is due to mice developing bone metastases on average 18 days after tumor cell inoculation. The lower metastases volume measured at T1 is therefore expected and in line with the previous study [6].

The previous measurements are global parameters in the complete ROI. To quantify the local features of blood vessels around metastases, we first measured the VMI and generated 3D mappings to describe the local distance from each point in the blood vessel compartment to the metastases compartment (Figure 7A,B). We then selected different VOIs (volume of interests: VMI < 300 µm, VMI < 700 µm, and VMI < 1000 µm) to quantify the local blood vessels. However, we found that the VOI (VMI < 300 µm) is too close to the metastases to extract enough blood vessel information to study. As for the VOI (VMI < 1000 µm), it might be more like global features rather than local features in some samples. Therefore, the VOI (VMI < 700 µm) was finally selected to investigate the local V.Th and VV/TV of blood vessels around metastases, as shown in Figure 7C–F.

The result showed that the local V.Th significantly decreased around the metastases in the T2C group compared to T2P (*p* < 0.05). As for the local VV/TV, there was a significant decrease (*p* < 0.05) in T2C. Globally, however, there is no significant change in VV/TV, as shown previously [7]. This indicates that the full blocking of the VEGF signaling pathway with the combination treatment has a stronger anti-angiogenic effect on the bone blood vasculature close to bone metastases.

## 4. Discussion

In this study, we proposed a new protocol to simultaneously segment and quantify 3D bone, blood vessels, and metastases in contrast-enhanced SR-µCT images.

The classification of bone and blood vessels from these kinds of images is difficult, due to partial volume effect, the similar contrast between bone and vessels, and intensity variations in the 3D SR-µCT images. In addition, the automatic segmentation of metastases is challenging since there is no direct contrast from the metastases in these SR-µCT images and they must be inferred from missing structure of the bone. Further, since computing costs might be relatively high for 3D images with a large size using classical segmentation methods, it is impossible to further investigate the relationship between metastases and vascularization using traditional image analysis methods.

To address these problems, we proposed to use a deep learning-based method (nnU-Net) with a minimal training data. The segmentation quality of bone (Dice = 0.99 (0.004)) and blood vessels (Dice = 0.97 (0.007)) obtained using the proposed method, as shown in (Figure 5), was improved compared to the segmentation (bone: Dice = 0.98 (0.003); blood vessels: Dice = 0.87 (0.027)) acquired using the previously proposed method [22]. Regarding the computing time, a deep learning method can achieve a faster prediction (30 s) compared to a traditional segmentation algorithm (previously proposed local phase asymmetry and watershed method, 360 s), under the same conditions (tested on an example volume of 700×700×100). In addition, we showed that the efficiency of generating the training data set was improved by correcting minor automatic segmentation errors in the blood vessels and bone microstructure. As for metastases, the manual annotations were not an extremely time consuming task, since we only require minimal training data (10 samples) to achieve a relatively high segmentation quality (Figure 5).

We found that the BV/TV (bone volume fraction) and V.Th (mean thickness of blood vessels) were significantly higher (*p* < 0.001 and *p* < 0.01, respectively) at T2 (time point: 22 days after breast cancer cell injection) than at T1 (time point: 8 days after breast cancer cell injection), as shown in (Figure 6A,C). This might be because mice were young growing animals. In contrast, when extracting VV/TV (blood vessel volume fraction), we observed that VV/TV was significantly lower (*p* < 0.01) at T2 than at T1 (Figure 6B). This was likely explained by the fact that TV increased faster compared to VV in the growing phase of mice. With regard to the Me.V/TV (metastases volume fraction), we found that the Me.V/TV significantly increased (*p* < 0.001) at T2 (Figure 6D). It has previously been reported that mice develop bone metastases on average 18 days after tumor cell inoculation [6]. Thus, T2 was a later phase in bone metastases development and more metastases were detected as expected.

The segmentation provided by the method allowed for the discovery that the combination treatment significantly affected local vascularization close to metastases. Globally, however, there was no significant change, as shown previously [7]. Specifically, to quantify the local features of blood vessels around metastases and provide more valuable information on the structure changes, we selected a VOI (VMI < 700µm) to measure V.Th and VV/TV, as shown in Figure 7C,D. In this case, we found that V.Th and VV/TV were significantly lower (*p* < 0.05) in the T2C group compared to T2P (Figure 7E,F). It is therefore highly likely that the combination treatment has a stronger anti-angiogenic effect on bone vasculature close to bone metastases.

## 5. Conclusions

A limitation of this study is that the investigation of the relationship between the blood vessels and metastases may require histological proof or analysis by a tumor pathologist. In particular, the histology method, which is a well-established research tool to analyze the undecalcified bone and vasculature using microscopy and staining, was not included in this study for comparison. Nevertheless, in the previous study, we have validated the proposed SR-µCT method with histology and histomorphometric analysis of bone tissue sections under the same animal model and drug treatments conditions, supporting the feasibility of the proposed method in this study [6]. Therefore, further studies can still take the preliminary results of this study into account and provide a better understanding of vasculature related bone metastatic processes. 

In conclusion, we showed for the first time that combining blood vessel contrast agents and SR-µCT imaging with a deep learning method (nnU-Net architecture) allows us to simultaneously segment 3D bone, blood vessels, and metastases, providing an unbiased method to quantitatively analyze the local features of blood vessels around metastases. We showed that the combined anti-angiogenic drug treatment significantly altered vascularization close to metastases compared to the placebo group in a mouse model of bone metastasis.

## Figures and Tables

**Figure 1 cancers-15-05609-f001:**
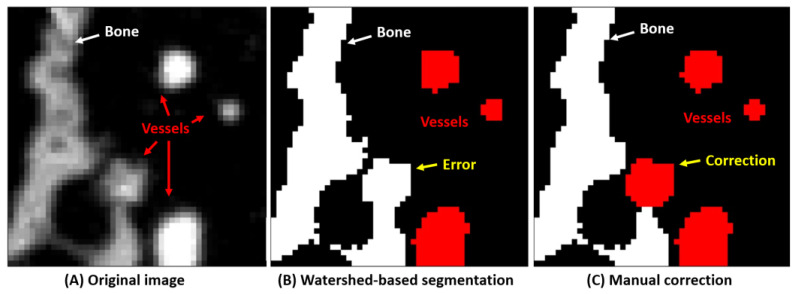
Training data set generation for bone and blood vessels. (**A**) the original image. (**B**) The image is first segmented using a watershed-based algorithm. (**C**) This segmentation is then improved manually to yield as good training data as possible.

**Figure 2 cancers-15-05609-f002:**
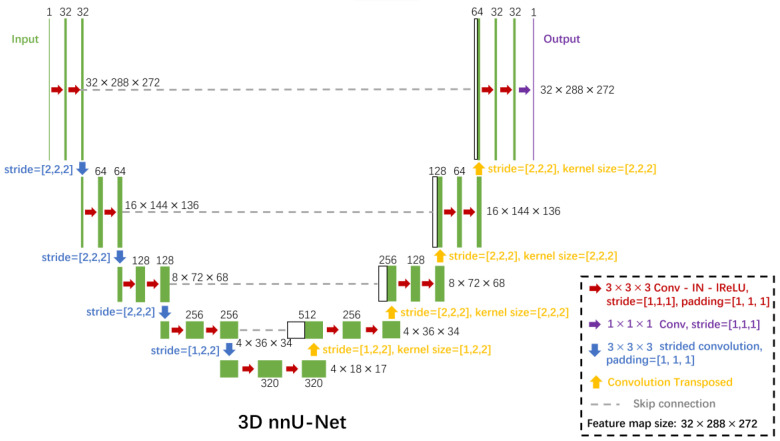
The nnU-Net convolutional neural network architecture for the segmentation of bone, vessels, and metastases from 3D SR-µCT images. Each green box corresponds to a multi-channel feature map. The number of channels is denoted on top of the box. The x-y-z size is provided at the right edge of the box. White boxes represent copied feature maps. The arrows denote the different operations. Conv indicates plain convolutions; IN, instance normalization; lReLU, leaky rectified linear unit.

**Figure 3 cancers-15-05609-f003:**
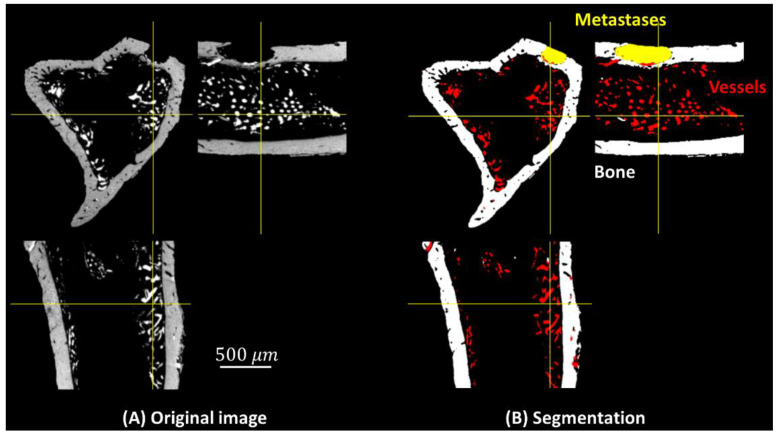
Three orthogonal slices in the 3D volume: (**A**) original image, (**B**) segmented image, bone (white), blood vessels (red), and metastases (yellow).

**Figure 4 cancers-15-05609-f004:**
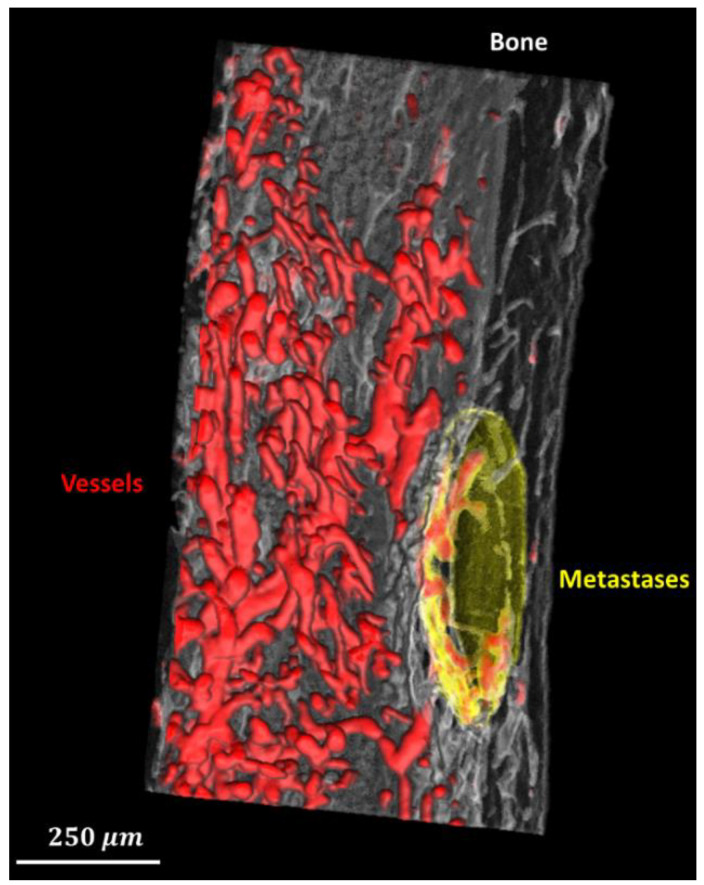
3D rendering of the segmented bone (translucent white), blood vessels (red), and metastases (translucent yellow).

**Figure 5 cancers-15-05609-f005:**
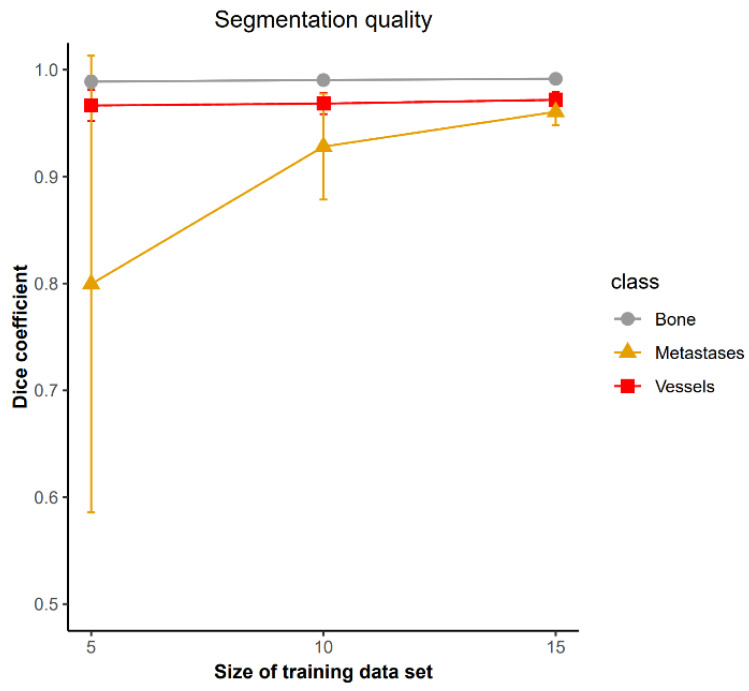
Evaluation of segmentation quality. Only a minimal training data set (10 samples) is necessary to achieve relatively high quality segmentation for bone, blood vessels, and metastases.

**Figure 6 cancers-15-05609-f006:**
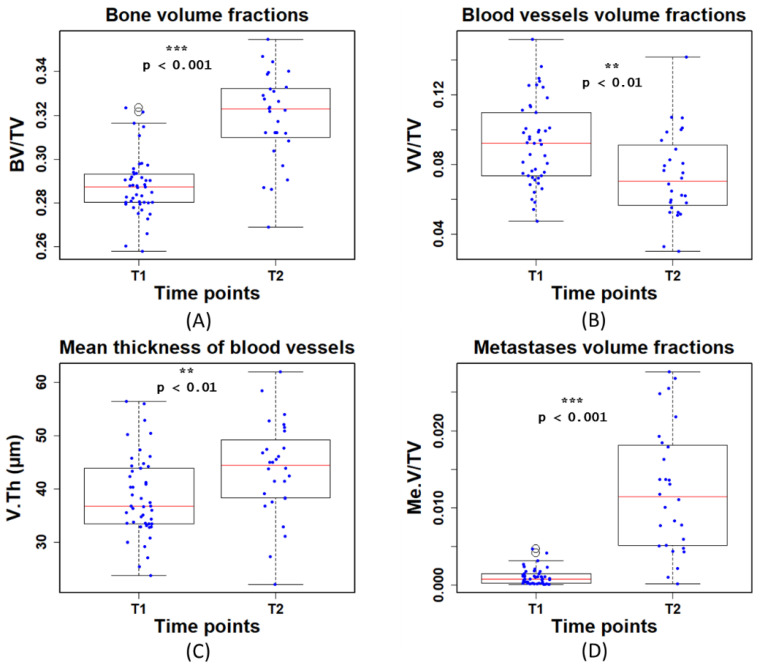
Quantitative measurements of **(A)** bone volume fraction (BV/TV), (**B**) blood vessel volume fraction (VV/TV), (**C**) mean thickness of blood vessels (V.Th), and (**D**) metastases volume fraction(Me.V/TV) at the first (T1) and second time point (T2). T1 and T2 denote time points at day 8 and 22 post tumor cell injection, respectively.

**Figure 7 cancers-15-05609-f007:**
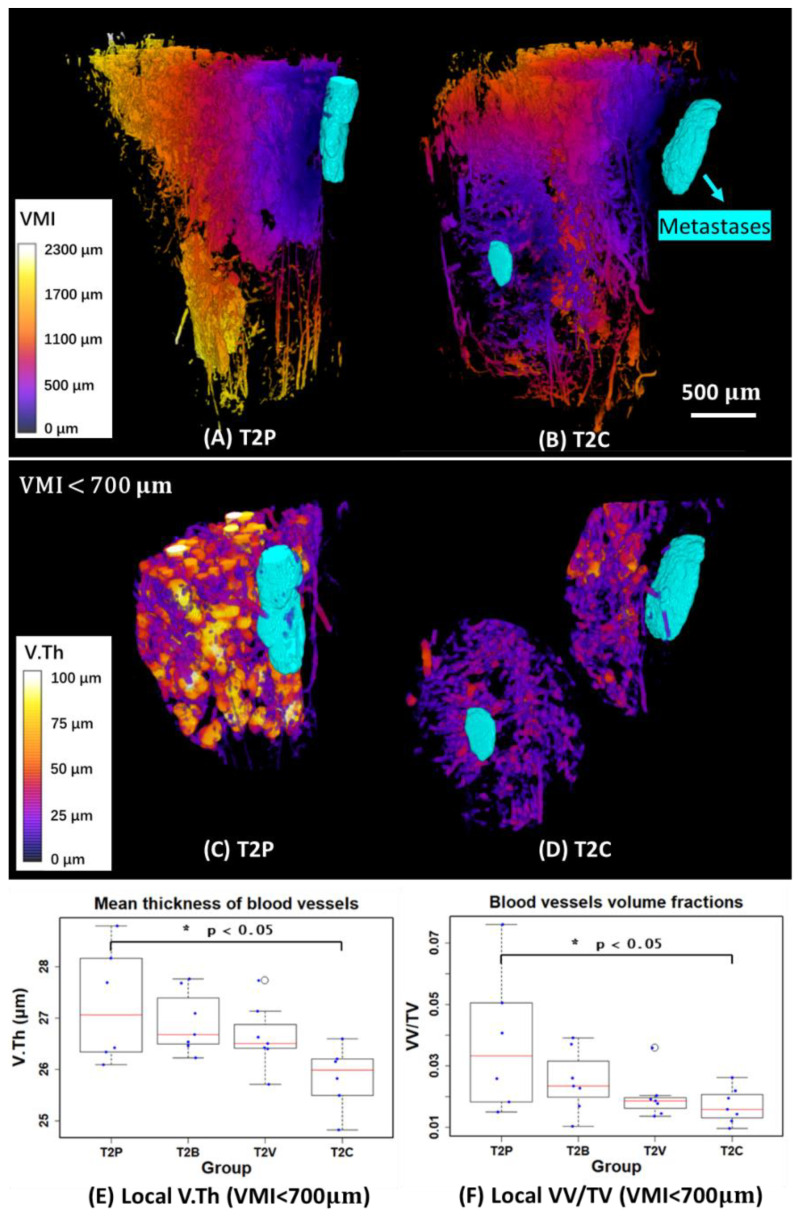
(**A**,**B**) 3D mappings of vascular metastases interdistance (VMI) and 3D rendering of metastases (in cyan) of two examples in the T2P and T2C groups. (**C**,**D**) 3D mappings of local vessel thickness (V.Th) and 3D rendering of metastases (in cyan) in VOIs restricted by 700 µm away from the metastases. (**E**,**F**) Quantitative measurements of local mean thickness of blood vessels (V.Th) and local vessel volume fraction (VV/TV) in VOIs restricted by 700 µm from the metastases. T1 and T2 denote the first and second time points, respectively. (P: Placebo, B: Bevacizumab, V: Vatalanib, C: Combination of Bevacizumab and Vatalanib).

## Data Availability

The data that support the findings of this study are available from the corresponding author upon reasonable request.

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
