# Peer review of "Quantitative Analysis of Bone, Blood Vessels, and Metastases in Mice Tibiae Using Synchrotron Radiation Micro-Computed Tomography"

_cancers, 2023, doi:10.3390/cancers15235609_

Round 1
Reviewer 1 Report
Comments and Suggestions for Authors
In the article entitled: “Simultaneous segmentation and quantitative analysis of bone, blood vessels, and metastases in mice tibiae using synchrotron radiation micro-computed tomography”, the authors describe an automatic method for segmentation of bone, blood vessels, and metastases in contrast-enhanced synchrotron radiation microcomputed tomography images.
The topic is very important since the method for automatic segmentation of the three structures seemed to better segment the three structures than previous methods discussed, and additionally, it does it faster. The article is well-written and delivers an overarching message throughout the different sections of the article. The flow of the paper is consistent and the different sections are clear and balanced. The figures in the paper are informative and clear. Some minor points that I would like the authors to clarify are the following:
1. In the Materials and Methods section, please indicate which software was used for each of the following sections:
· 2.4 Evaluation criteria
· 2.5 Quantitative parameter extraction
· 2.6 Statistical analysis
2. In the results section please indicate which software was used for 3D volume rendering (section 3.1. qualitative results of segmentation).
3. In the results section 3.3 Quantitative analysis, the authors say at the end of the first paragraph that: “The lower metastases volume measured at T1 is therefore expected and in line with our previous work (Xu et al., 2022)”. Can the authors compare deeply the results obtained in both works? Since it is expected to be better as segmentation, I would expect an improvement in the quantification. Does it really make a difference?
4. Throughout the manuscript, it appears at the bottom the following information: Cancers 2022, 14, x, please correct this since it seems that might be a typo error coming from the previous published paper.
5. In the introduction section, the authors state that: “To image vascularization within the whole bone, 3D X-ray micro-computed tomography (μCT) can be used. It is a powerful tool to image bone and blood vessels using contrast agents (Moore et al., 2003; Zhang et al., 2005). However, 3D X-ray μCT usually requires bone decalcification to visualize the vascular architecture. Thus, bone and blood vessels cannot be analyzed simultaneously”.
However, recent work from Dr. Ruslan Hlushchuk and his team at the University of Bern showed that it is possible to visualize simultaneously bone and blood vessels using contrast agents ( https://www.biorxiv.org/content/10.1101/2023.03.08.531678v2)
Author Response
Reviewer 1
In the article entitled: “Simultaneous segmentation and quantitative analysis of bone, blood vessels, and metastases in mice tibiae using synchrotron radiation micro-computed tomography”, the authors describe an automatic method for segmentation of bone, blood vessels, and metastases in contrast-enhanced synchrotron radiation microcomputed tomography images.
The topic is very important since the method for automatic segmentation of the three structures seemed to better segment the three structures than previous methods discussed, and additionally, it does it faster. The article is well-written and delivers an overarching message throughout the different sections of the article. The flow of the paper is consistent and the different sections are clear and balanced. The figures in the paper are informative and clear. Some minor points that I would like the authors to clarify are the following:
- In the Materials and Methods section, please indicate which software was used for each of the following sections:
- 2.4 Evaluation criteria
According to the comment, we have added text: “To calculate the Dice coefficient, we used the open source RStudio software with programming language R.”
- 2.5 Quantitative parameter extraction
According to the comment, we have added text: “The parameters were calculated in a shell script with the Insight Toolkit (ITK), which is an open source library for image analysis.”
- 2.6 Statistical analysis
According to the comment, we have added text: ”Statistical analysis was implemented using RStudio software.”
- In the results section please indicate which software was used for 3D volume rendering (section 3.1. qualitative results of segmentation).
According to the comment, we have added text: “We used the open source image processing software Fiji/ImageJ to perform the 3D volume rendering.“
- In the results section 3.3 Quantitative analysis, the authors say at the end of the first paragraph that: “The lower metastases volume measured at T1 is therefore expected and in line with our previous work (Xu et al., 2022)”. Can the authors compare deeply the results obtained in both works? Since it is expected to be better as segmentation, I would expect an improvement in the quantification. Does it really make a difference?
In our previous work, we showed manually estimated the contour of the metastatic lesions on trabecular bone (Xu et al., 2022). However, in the current study, we automatically segmented metastases on cortical bone. Thanks to the reviewer's reminder, we found that it is not appropriate to cite our previous work (Xu et al., 2022) here, since the segmentation target is different from this study. Finally, we update a new reference paper “Bachelier et al., 2014”, which reported that bone metastases are often developed in 18 days after tumor cell inoculation. In our study, two time points of 8 days (T1) and 22 days (T2) after tumor cell injection were set up.
The updated sentence is “The lower metastases volume measured at T1 is therefore expected and in line with the previous study (Bachelier et al., 2014)”.
Bachelier, R. et al. (2014) ‘Combination of anti‐angiogenic therapies reduces osteolysis and tumor burden in experimental breast cancer bone metastasis’, International Journal of Cancer.
- Throughout the manuscript, it appears at the bottom the following information: Cancers 2022, 14, x, please correct this since it seems that might be a typo error coming from the previous published paper.
According to the comment, we have corrected this error in the manuscript.
- In the introduction section, the authors state that: “To image vascularization within the whole bone, 3D X-ray micro-computed tomography (μCT) can be used. It is a powerful tool to image bone and blood vessels using contrast agents (Moore et al., 2003; Zhang et al., 2005). However, 3D X-ray μCT usually requires bone decalcification to visualize the vascular architecture. Thus, bone and blood vessels cannot be analyzed simultaneously”.
However, recent work from Dr. Ruslan Hlushchuk and his team at the University of Bern showed that it is possible to visualize simultaneously bone and blood vessels using contrast agents ( https://www.biorxiv.org/content/10.1101/2023.03.08.531678v2)
According to the comment of reviewer, we have replaced the sentence “Thus, bone and blood vessels cannot be analyzed simultaneously” with: “Until recently, a research showed that it is possible to simultaneously visualize bone and blood vessels without decalcification using a polymer-based contrast agent µAngiofil and 3D X-ray μCT (D. Haberthür et al. 2023)”
- Haberthür et al., ‘MicroCT-based imaging of microvasculature within the bone tissue’, Physiology, preprint, Mar. 2023. doi: 10.1101/2023.03.08.531678.

Reviewer 2 Report
Comments and Suggestions for Authors
The manuscript by Hao Xu and Max Langer presents a technical follow-up to several related papers that utilize Grenoble's synchrotron and machine learning in a bone metastasis model in mice. The authors introduce an innovative method and discuss its application in testing drug responses within the mouse model. While the reviewer finds the work publishable, there are some minor issues that could be addressed for improvement.
- The title is rather lengthy and should be shortened.
- In line 38, add a space in front of the citation bracket, and check all other citations for consistency.
- In line 109, briefly explain or spell out the abbreviation "VOI" for clarification, ensuring readers understand it does not come with a unit.
- Sharing the data for re-analysis, following FAIR principles, could benefit the scientific community. See for reference: “Community standards for open cell migration data” https://doi.org/10.1093/gigascience/giaa041
- In lines 56-64, consider rephrasing to clarify for the reader what synchrotron radiation is. Is it simply another form of X-rays, and is it only available in Grenoble, not in any CT?
- In lines 95-110, for better clarity, mention briefly that tibiae specimens were collected from perfused animals, i.e., not living animals (as some readers may associate with Computed Tomography). Specify whether or not animals were perfused with fixing agents like formalin or glutaraldehyde. If not, provide information on the time elapsed between perfusion and synchrotron analysis, addressing potential degradation processes in unfixed tissue leading to artifacts and osmotic changes.
- In lines 248-309, consider expanding the conclusions to include more limitations of the study. For example, a) the study does not definitively demonstrate that changes in structure result from metastases, as there is no histological proof or analysis by a tumor pathologist. The structures observed may theoretically stem from immune reactions or other factors. b) Acknowledge that standard histological methods, applied for decades also on tumor vasculatisation, may yield comparable or superior results and explain why these methods were not included for comparison.
Author Response
Reviewer 2
The manuscript by Hao Xu and Max Langer presents a technical follow-up to several related papers that utilize Grenoble's synchrotron and machine learning in a bone metastasis model in mice. The authors introduce an innovative method and discuss its application in testing drug responses within the mouse model. While the reviewer finds the work publishable, there are some minor issues that could be addressed for improvement.
1. The title is rather lengthy and should be shortened.
The title was updated as: “Quantitative analysis of bone, blood vessels, and metastases in mice tibiae using synchrotron radiation micro-computed tomography”
2. In line 38, add a space in front of the citation bracket, and check all other citations for consistency.
According to the comment, a space has been added in front of the citation bracket throughout the manuscript.
3. In line 109, briefly explain or spell out the abbreviation "VOI" for clarification, ensuring readers understand it does not come with a unit.
“VOI: volume of interest” has been mentioned in the manuscript.
4. Sharing the data for re-analysis, following FAIR principles, could benefit the scientific community. See for reference: “Community standards for open cell migration data” https://doi.org/10.1093/gigascience/giaa041
The data are available from the corresponding author upon reasonable request.
5. In lines 56-64, consider rephrasing to clarify for the reader what synchrotron radiation is. Is it simply another form of X-rays, and is it only available in Grenoble, not in any CT?
According to the comment, we have added text below:
“Synchrotron radiation is an emitted electromagnetic radiation in a wide spectral range from infrared to hard X-rays. Synchrotron radiation X-ray beam is characterized by the high flux and high degree of mono-chromaticity, compared to the standard X-ray. Electron storage rings are used to provide synchrotron radiation in a dedicated facilities around world, such as ESRF(Europe), ALS(America), Spring-8(Japan), SSRF(China).”
6. In lines 95-110, for better clarity, mention briefly that tibiae specimens were collected from perfused animals, i.e., not living animals (as some readers may associate with Computed Tomography). Specify whether or not animals were perfused with fixing agents like formalin or glutaraldehyde. If not, provide information on the time elapsed between perfusion and synchrotron analysis, addressing potential degradation processes in unfixed tissue leading to artifacts and osmotic changes.
According to the comment, we have added text below:
“The mice were sacrificed by anesthetic overdose and injected with a contrast agent (barium sulfate) for vascular imaging. Afterwards, tibiae were dissected, fixed in paraformaldehyde, embedded in methylmethacrylate, and kept for SR-µCT imaging. “
7. In lines 248-309, consider expanding the conclusions to include more limitations of the study. For example, a) the study does not definitively demonstrate that changes in structure result from metastases, as there is no histological proof or analysis by a tumor pathologist. The structures observed may theoretically stem from immune reactions or other factors. b) Acknowledge that standard histological methods, applied for decades also on tumor vasculatisation, may yield comparable or superior results and explain why these methods were not included for comparison.
According to the comment, we have added text below in the conclusion section:
“A limitation of this study is that the investigation of relationship between the blood vessels and metastases may require histological proof or analysis by a tumor pathologist. Especially, the histology method, which is a well-established research tool to analyze the undecalcified bone and vasculature using microscopy and staining, was not included in this study for comparison. Nevertheless, in the previous studies, we have validated the proposed SR-µCT method with histology and histomorphometric analysis of bone tissue sections under the same conditions of animal model and drug treatments, supporting the feasibility of the proposed method in this study (Bachelier, R. et al. 2014). Therefore, further study still can take the preliminary results of this study into account and provide a better understanding of vasculature related bone metastatic processes.”
Bachelier, R. et al. (2014) ‘Combination of anti‐angiogenic therapies reduces osteolysis and tumor burden in experimental breast cancer bone metastasis’, International Journal of Cancer.

Reviewer 3 Report
Comments and Suggestions for Authors
Cancers-2691989
Simultaneous segmentation and quantitative analysis of bone, blood vessels, and metastases in mice tibiae using synchrotron radiation micro-computed tomography by Xu and Langer propose a new method for simultaneous segmentation of bone, blood vessels and metastases from contrast enhanced 3D SR-µCT images based on the nnU-Net architecture. In the current study n this study, the authors showed that only minimal training data was required to achieve high quality of segmentation.
Comments:
1. Training data set generation for bone and blood vessels Fig 1 (A) original image is not clear and it will be nice to see non-digitized image.
2. Is that quantify the metastatic tumor volume in Fig 4
Author Response
Reviewer 3
Cancers-2691989
Simultaneous segmentation and quantitative analysis of bone, blood vessels, and metastases in mice tibiae using synchrotron radiation micro-computed tomography by Xu and Langer propose a new method for simultaneous segmentation of bone, blood vessels and metastases from contrast enhanced 3D SR-µCT images based on the nnU-Net architecture. In the current study n this study, the authors showed that only minimal training data was required to achieve high quality of segmentation.
Comments:
1.Training data set generation for bone and blood vessels Fig 1 (A) original image is not clear and it will be nice to see non-digitized image.
In this study, to clearly show the details of bone and blood vessels, it would be good to zoom in on the images. However, due to the small size of object structures (thin bone and blood vessels) and limitation of image resolution, the small objects were presented with a small number of voxels, leading to “digitized” image as Fig 1 (A). Thus, it may be difficult to avoid using “digitized” original image.
2.Is that quantify the metastatic tumor volume in Fig 4
We qualitatively show 3D rendering of the segmented metastases in Fig. 4 and quantitatively show the metastases volume fraction (Me.V/TV) in Fig. 6 (D).

Round 2
Reviewer 3 Report
Comments and Suggestions for Authors
The quality of the manuscript is improved